# Effects of gender sensitive language in job listings: A study on real-life user interaction

**Dominik Hetjens**[1]*, **Stefan Hartmann**[2]

**1** Institute of German Studies and Media Cultures, Technische Universität Dresden, Dresden, Sachsen, Germany, **2** Department of German Studies, Heinrich-Heine-Universität Düsseldorf, Düsseldorf, Nordrhein-Westfalen, Germany

\* dominik.hetjens@tu-dresden.de

## Abstract

The possible impact of gender-sensitive language on readers is among the most controversially debated issues in linguistics and beyond. Previous studies have suggested that there is an effect of gender-sensitive language on mental representations, based on data gathered in laboratory settings with small groups of participants. We add a new perspective by examining correlations of authentic language use with authentic user interaction on a recruitment website. Drawing upon a large dataset provided by the recruitment platform Step-Stone, we evaluate whether job advertisements using certain kinds of gender-sensitive language in their titles correlate with higher proportions of views by female users. Our results indicate that there are differing effects depending on the type of gender-sensitive language that is used. Overall, the strongest correlation can be found with terms that include the feminine suffix *-in*.

**Data Availability Statement:** The data can be found in a sheet here: https://uni-duesseldorf.sciebo.de/s/pKu2B1Np4wf1yqw.

## 1. Introduction

Gender-sensitive language has become a controversial topic across many different languages and cultures [1, 2]. It has been introduced with the intention to reduce stereotyping and discrimination. A prominent example is the push to replace certain job titles in English, e.g. *police offer* instead of *policeman* or *fire fighter* instead of *fireman* [2 p2].

This can be seen as a response to the observation that certain types of gender asymmetry that can be observed in society are reflected in language. Research has indeed shown that a male bias, i.e. "an implicit assumption that an undefined person is a man" [3 p110], is pervasive across languages. In English, to mention a prominent example, attempts to reduce this bias have led to an increase in the use of singular *they* instead of the masculine pronoun as an 'unmarked' default form [4], sparking language-ideological debates that share similarities with those in the German-speaking world that will be the focus of the present paper [5].

Here, the issue is more complex, since German belongs to a group of languages called 'gender languages'. In the case of nouns referring to animates, especially persons [6 pp5–6], these languages show a correspondence between 'feminine' and 'masculine' grammatical gender on the one hand and biological and/or social gender on the other. As a result, some have argued that sexist ideologies are deeply ingrained in the grammars of languages, especially in their

**Funding:** The author(s) received no specific funding for this work.

**Competing interests:** The authors have declared that no competing interests exist.

grammatical gender systems [7]. Hence, the proposed gender-sensitive forms in German are not limited to a small number of specific job titles as in the case of *policeman* or *fireman*, but affect the majority of person nouns, among them most job titles. The use of these forms, referred to in German as *gendern* 'gendering', is the topic of passionate debates and political campaigns [8 p210, 9]. Both sides of the debate appeal to linguistics to substantiate their claims. While some of these claims are based on moral or political ideas, others are rooted in linguistic concepts [9 p106]. Positions of linguists on the topic vary greatly, depending not only on their specific area of expertise, but also on certain basic assumptions about (German) grammar, the role of linguistics in society, and their views regarding the validity of the research methods typically used in empirical approaches to gender-sensitive language.

When it comes to the question of whether the use of gender-sensitive language has a real, measurable effect, the study of job listings can be considered of particular relevance, as the positions that are advertised relate to social and economic hierarchies. As such, online job listings, and data about how users interact with them, can be a promising resource for investigating whether and how the use of gender-sensitive language might impact potentially far-reaching professional decisions, especially when combined with demographic metadata. This is why this paper sets out to explore how different types of gender-sensitive language used in the titles of online job advertisements correlate with the relative numbers of views by female users. In doing so, we present, to the best of our knowledge, the first study on this topic that draws on a large-scale analysis of authentic user interaction data, rather than on data obtained in laboratory settings. We argue that this can offer new perspectives on the topic and help to address some of the issues that have been raised with regard to previous empirical research on gender-sensitive language.

In section 2, we will provide a brief overview of grammatical genders in German and the use of masculine forms for mixed groups and abstract entities (the so-called generic masculine form), which is at the centre of the criticism put forward by feminist linguistics. We will then summarise the current state of the art, roughly categorising previous empirical research on gender-sensitive language into corpus-based and experimental approaches. We will particularly focus on previous research not only in linguistics but also in the social sciences that addresses the effects of different types of gender-sensitive language on the perception of job listings (section 3). We then turn to our own case study, in which we analyse user interaction data provided by the online recruitment agency StepStone (section 4). In section 5, we offer an in-depth discussion of the results; section 6 summarises and concludes the paper.

Before we begin, several terminological notes are in order: Firstly, in analogy to the German term *gendern*, we will use the term *gendering* in this paper to refer to the use of gender-sensitive language. Likewise, we also use the verb *to gender* in the sense of 'using gender-sensitive variants of a specific term'. Secondly, we use the widespread term *generic masculine* (German: *generisches Maskulinum*, sometimes also referred to as *masculine generic*, see e.g. [10]) to refer to uses of masculine nouns that include both male and female referents, as in *alle Nobelpreisträger* 'all Nobel laureates'. Kotthoff and Nübling [11] point out that the use of the term *generic* in this context deviates from its use in other domains of linguistics, which is why some researchers speak of the 'so-called' generic masculine instead [11 p91], or use other terms such as "geschlechtsübergreifendes Maskulinum" (roughly: 'gender-spanning masculine', see e.g. [12]). While the latter term is arguably the most accurate one, there seems to be no fully adequate English equivalent, which is why we stick with *generic masculine* for the remainder of this paper. Lastly, we will use the term *gender-sensitive language*, not the possibly more common *gender-fair language*, to avoid any inherent evaluation with regard to the moral or functional quality of the linguistic forms referred to.

## 2. Overview of grammatical genders and the 'generic masculine' form

In German, each noun belongs to one of three classes (grammatical genders): *Der Mond* 'the moon' (masculine), *die Sonne* 'the sun' (feminine), *das Boot* 'the boat' (neuter). When it comes to nouns referring to persons, it is common in the German-language literature to distinguish between *Genus*, i.e. grammatical gender, on the one hand, and *Sexus*, i.e. biological sex and/or social gender, on the other [e.g. 13]. Hellinger and Bußmann [6 pp6-8] make a more fine-grained distinction between *grammatical gender*, *lexical gender* and *referential gender*. Grammatical gender refers to an inherent property of nouns that is, in principle, independent of the actual gender of the referent, e.g. German *der Mond* 'the moon' (masculine), French *la lune* 'the moon' (feminine). Lexical gender (also called *semantic gender)* refers to semantic properties like [male] or [female] that are present in some nouns (e.g. *aunt* [f.] vs. *uncle* [m.]), while referential gender is conceived of as an extra-linguistic phenomenon based on social or biological categorisations. We will adopt these terms here, especially because *semantic gender* seems much more appropriate a term than *sexus*, as the latter strongly evokes the notion of biological sex, while gender as a social construct is at least as important to the questions at hand as biological features [6].

As mentioned before, there is considerable correspondence between grammatical and referential gender in nouns referring to persons [6 p7], especially in the singular, e.g. *der Mann* 'the man', *die Frau* 'the woman'. Importantly, most role nouns, which are by default masculine in German, can be turned to feminine nouns by adding the female suffix *-in*, e.g. *Linguist* 'linguist' > *Linguistin* 'female linguist'. This is especially relevant in the case of job titles, which are the main topic of the present study. Role nouns usually exist in pairs [e.g. 14 p291], with the female form being marked by the suffix *-in*. In addition, many of them are agent nouns with the agentive suffix *-er*, resulting in the stereotypical pattern *der Lehrer* 'the teacher (m.)'– *die Lehrerin* 'the teacher' (f.), with the plural forms being *die Lehrer* and *die Lehrerinnen* [14 p293].

Thus, in many cases when referring to groups of mixed genders or abstract entities (e.g. 'a teacher *per se*'), there is no (grammatically) 'neutral' way to do so. The traditional use of the male version (*die Lehrer*), the so-called generic masculine form (subsequently referred to as *GM*), has been criticised since the advent of feminist linguistics in the 1970s. It was argued that German tends to make women 'invisible', which could potentially entail social consequences [2, 15]. On this view, there is a link between grammatical and semantic gender, rooted in the concepts elicited by a word, especially if the word in the singular is commonly used to refer to a certain referential gender (also see [16] for an overview): for instance, if *Anwalt* 'lawyer' or its plural form *Anwälte* 'lawyers' are frequently used to refer to male individuals, the term will gradually become associated with male practitioners of this profession. For this reason, many alternatives to the masculine form have been suggested over the last decades. Different types of 'gendering' have become common in German, including the use of both the masculine and the feminine form (*Linguistinnen und Linguisten*) and the combination of male and female forms using various types of morpheme separators (often pronounced as a glottal stop in spoken language) such as *Linguist\*innen*, *Linguist_innen*, *Linguist:innen*, or using the capital I as in *LinguistInnen* [16 p119].

## 3. State of the art: Corpus-based and experimental approaches to generic masculines and alternative forms in German

Current research on whether and to what extent women are represented in the GM shows no consensus. Some of the controversy in the literature is based on differences in the methods

and fundamental assumptions of different linguistic approaches. Among linguists who study the language system mostly independent from language use, there is a tendency to argue that *Genus* (grammatical gender) and *Sexus* (biological sex or social gender) are largely independent categories, to the extent that it has been suggested to rename grammatical genders 'noun classes' to avoid any association with biological or social gender [17 p41]. It has also been argued that generic masculine forms are a case of 'auto-hyponymy,' where a grammatically masculine noun like *Bäcker* 'baker' is a hyperonym of both *Bäcker* 'male baker' and *Bäckerin* 'female baker' [18 p73]. While this approach features prominently in public discourse, it has been criticised for its neglect of empirical evidence, which is gathered mostly in the context of corpus-based and experimental approaches. We will describe these empirical approaches more thoroughly, as they are central for our own study.

Corpus-based approaches have focused on three issues: the diachronic frequency development of GM forms and their alternatives, the sociolinguistic implications of gender-sensitive forms, and their semantics as gauged via distributional methods. Starting with the first-mentioned line of research, corpus studies have clearly shown a general increase in the use of alternatives to the GM, indicating a rising need for, and acceptance of, these forms [16 p138, 19 p11]. However, the GM still appears to be by far the most widely-used form to refer to groups of mixed genders, except in specific academic and administrative texts, in which different types of gender-sensitive language are applied [8 p216, 16 p139]. As for the question of how old the GM actually is, there are conflicting results. In response to Diewald's [14] claim that the GM is a fairly young convention, Trutkowski and Weiß [20 p35], based on an analysis of historical corpus data, argue that the GM "has always been part of German grammar" [20 p35]. In a study of the historical and synchronic use of generic masculines in predicative constructions, Kopf [21], by contrast, shows that generic uses of masculine nouns are the exception.

While the question of whether or not the GM is a long-established convention plays a considerable role in public discourse, its relevance for understanding the use and processing of masculine forms in present-day language is arguably negligible. What is more relevant are the sociolinguistic and semantic implications of the forms under scrutiny. As mentioned above, there are numerous alternatives to GMs [19], but one of them, the asterisk form as in *Linguist*innen*, tends to be singled out in public discourse. Focusing on this form, Sökefeld [16] shows that there is considerable variation in the use of the GM and alternative forms within and between texts, and argues that the asterisk serves the purpose of signaling a metalinguistic awareness and a feminist position. This is in line with Kotthoff's [9 pp105,116] assumption that the asterisk is used to connect its users to a particular "socio-symbolic cosmos". We will return to these ideas in the discussion of our own findings (section 5).

Turning to the semantics of GM and alternative forms, Schmitz et al. [10] use distributional semantics to investigate the meaning of (supposedly) generic masculines, explicit masculine forms, and explicit feminine forms. They use a semantic vector-space approach, which gauges semantic similarities and differences between words by comparing the contexts in which they occur in a quantitative way. Their results show that masculine generic and specific forms behave very similarly both in the singular and in the plural, while explicitly feminine forms are found to be significantly different to both generic *and* explicit masculines, which leads the authors to the conclusion that there is "no genericity in sight".

These diverging conclusions show that while corpus-based approaches are important for gauging the extent to which generic masculines as well as different types of gender-sensitive language occur in actual language use, they face various challenges. Most importantly, it is often not possible to disentangle semantic and referential gender, which is why much research

on the question of how the GM is actually *understood* in present-day language draws on behavioural experiments instead.

Beginning in the late 1980s, researchers have made efforts to find out how the GM is understood by readers and which associations it may evoke. Kotthoff and Nübling [11 p99] argue that this is crucial in evaluating whether the GM might be problematic, as argued for in feminist linguistics, because successful communication is based more on recipients' actual understanding of messages than on the expressed intention of authors. This is why different recipient-centered experiments have been conducted, using reaction time tests, blank-filling exercises, eye tracking and acceptability tests, among other methods (for an overview see [11 p115, 22–26]). While individual findings vary to some extent, all studies suggest that the masculine form is more likely to elicit mental representations of men than of women, creating a male bias [24 p553].

In a much-cited study, Gygax et al. [25] have combined reaction time testing and acceptability testing, comparing the understanding of role nouns like *spies* or *football players* in German, French and English. The participants' task in the German and French tests was to judge whether a sentence using an anaphoric NP identifying the group as male or female was an acceptable continuation of a preceding sentence using a (generically intended) form, which was grammatically masculine in the German and French stimuli. For example, the participants should judge whether the sentence *The social workers were walking through the station.* could be sensibly continued with *Since sunny weather was forecast several of the women/the men weren't wearing a coat* [25 p472]. Their results show that in German and French, continuations with 'the men' are both deemed more acceptable than with 'the women' and lead to significantly lower reaction times. Thus, they conclude that their results indicate a "very strong effect [. . .], biasing the participants' mental gender representation towards a male representation" [25 p478].

Schunack and Binanzer [27] take the same approach but extend the study design by taking newer, non-binary forms such as *Lehrer*innen* into account. Interestingly, their results show that the capital I form *LehrerInnen* leads to higher estimates for the proportion of women in inherently female-biased nouns, while the asterisk form shows the biggest increase in women estimates for male-biased nouns and a much smaller increase for neutral and female-biased nouns.

However, some of these experimental studies have been criticised for their small sample size, theoretical assumptions, and overall design [11 p108, 20 p15]. Trutkowski and Weiß argue that the effects observed in all these studies are largely syntactic in nature [20 p15], and they suggest that the perceived mismatch is less likely to be rooted in actual mental representations of participants, but rather in incongruent syntactic components. More generally, Trutkowski and Weiß also argue against the "psychologistic" concept of meaning that, in their view, underlies these studies: According to them, the associations that these experiments tap into are "subjective, private and eventually irrelevant to the meaning of a word". This shows strikingly that linguists' positions on the topic of gender-sensitive language depend to a considerable extent on certain basic assumptions that tend to differ among linguistic sub-disciplines (for an overview and a characterisation of some of these basic assumptions, see [11 pp99,115]).

Regardless of whether and in what sense the extent to which a person noun evokes female representations can be considered part of its meaning, a crucial question that goes beyond a purely linguistic perspective is whether and how the use of the GM vs. alternative forms may entail actual social consequences. This is why several studies have set out to explore how the wording of job advertisements might influence interest in positions by potential female applicants. In a publication summarising three smaller experiments among psychology students,

Gaucher et al. [1] conclude that women report less interest in a job position after reading English job advertisements including words that had been found to be commonly associated with male stereotypes, e.g. *determined*, *individualistic*, *superior*. When it comes specifically to gender-sensitive variants in German and job positions, there are only a few relevant studies. Horvath and Sczesny [28] conducted a hiring simulation experiment, showing potential employers a text advertising a position as well as CVs of potential candidates. Their results suggest that women who apply for a job are less likely to be perceived as fitting a position if the text advertising the position uses the GM.

While this might give some insight into how the GM could have an influence in the hiring process, it does not tell us much about the effect on applicants or other readers of advertisements. This is what has been tested in a comparatively large experiment (N = 591) with primary school students [29, 30]. This study found that girls were less likely to state that they felt competent to do a specific job if the job description used only the masculine form.

While Horvath & Sczesny's study compares the use of the GM to explicit mentions of both masculine and feminine forms, more recent studies also take other gendering types into account. As Körner et al. [24] mention, the asterisk (as in *Lehrer\*innen*) has become the most common form of gendering using morpheme-separating symbols. In an experiment based on Gygax et al. [25], they compare the effects of the asterisk to that of the GM. They conclude that the use of the "gender star" results in a female bias that is measurable in acceptability tasks, whereas the GM results in a stronger male bias, even though the generic intention was communicated explicitly in the experiment.

However, all these studies are based on experimental settings and relatively small groups of participants. One exception to this is a data-driven analysis focussing on the effect of gendered wording on the overall popularity of job advertisements by the Swiss job agency Jobchannel [31]. However, this study does not account for the gender of users, which is why it does not make any statements about whether women might be more likely to apply for a job if the job title avoids the use of the GM. This distinction is important, because any observed effect on the views of job advertisements might be caused by a number of extralinguistic factors not connected to the gender of the recipients. Still, the study yielded a striking result: Among 280,000 job titles, those using gender-sensitive language were viewed considerably more frequently in total than those that didn't. Among them, the most successful ones were job titles that used a slash separating the suffix indicating the female gender: *Lehrer/in* 'teacher'. Almost as successful, according to this study, were forms including the asterisk or the colon: *Lehrer\*in* or *Lehrer: in*, whereas the GM appears to consistently deliver below-average results. Another finding of the study is that the effect varies between different job sectors, which is why the authors recommend that employers should use different types of gendering, depending on the respective job sectors [31 p18]. However, the study does not seem to account for individual job positions that might have a strong influence on the dataset for each job sector, e.g. predominant jobs in a sector for which there is a typical term that is not specified in regard to its semantic gender (see the discussion in section 5 below).

The observed effects can be linked to a variety of possible factors. As Kotthoff [9 p106] suggests, gendering might be used to form a group identity around the usage of a certain type of language. Drawing on concepts from interactional sociolinguistics and anthropological linguistics, she describes different styles of gendering in their relation to practices of metapragmatic positioning [9 p121]. In this light, it seems reasonable to assume that effects of gendering in job titles might, at least in part, be caused not by grammatical genders evoking certain mental representations, but by assumptions about the ideological character of the respective companies and their will to position themselves by using a certain type of language.

Summing up the findings from corpus-based and experimental approaches, the question of whether masculine nouns can be interpreted in a generic sense has been addressed from a variety of perspectives, some of which differ quite drastically regarding their underlying assumptions. The more theoretically-oriented systemic-linguistic approach argues that masculine person nouns have both a generic and a specific reading, the former including referents of all genders, the latter only referring to male persons. It is an empirical fact that masculine nouns *can* be and are being used for referring to persons of all genders, as corpus-based approaches have shown. It is therefore arguably an exaggeration to call the generic masculine a "fiction" (e.g. [13] p44, 15]). However, there is ample evidence both from corpus linguistics and from experimental approaches that generic masculines are much less common in actual language use, both synchronically and historically, than assumed by some proponents of the systemic approach, and that GM forms tend to evoke male-biased representations.

Importantly, the different empirical approaches that have been used to address these questions all come with their own problems and limitations. And given the complexity of the issues at hand, it would in virtually all cases be too simplistic to postulate a direct monocausal relationship between the use of the GM or certain gendering types on the one hand and the behavioural observations obtained in the different studies on the other. As such, we can, in most if not all cases, conceive of possible alternative scenarios unrelated to mental representations that can explain (some of) the effects observed in the studies reviewed above. For one thing, we have seen that in corpus-based studies it is not always possible to tell semantic and referential gender apart. In a similar vein, grammatical and semantic factors cannot always be clearly told apart in experimental studies, which is why e.g. the mismatch between nouns and anaphoric pronouns in Gygax et al.'s [25] stimuli sentences could be interpreted as having to do more with grammatical agreement than with semantic factors. Furthermore, we cannot fully exclude the possibility that in experimental settings, participants are biased in one way or another. For example, when asked to estimate the proportion of women in a group, participants might guess the goal of the study, which in turn might influence their responses. Another closely related issue is that the samples of participants that experimental studies draw on are often small and non-representative, mostly consisting of students, who can be expected to be highly aware of the meta-linguistic and socio-political discussions about language and gender.

Drawing on authentic user interaction data arguably eliminates some of these problems and offers a more direct window to behavioural correlates of different types of gendering, which ideally allows for drawing more reliable conclusions about how language users interpret nouns that use either a generically masculine form or different types of gendering. This is the research gap that we would like to fill with the case study to which we now turn.

## 4. Case study

### 4.1 Aim, data and methods

Analysing user interaction on websites allows for combining the advantages of corpus-based and experimental approaches, while avoiding some of the specific disadvantages associated with laboratory settings. Unlike the stimuli that have been used in behavioural experiments, our approach draws on authentic texts, i.e. examples of real-world language, and the recorded user interactions are interactions that happened on the website, outside of any experimental context.

We draw on a dataset provided by the recruitment platform StepStone. On their website, users can search for specific jobs, which yields a list of job titles. By clicking on the respective title, users can view the full job advertisement. Our main goal is to see whether specific gendering types in job listings correlate with a higher number of views by users who identify as

female. We draw on a dataset containing German-language job listings from the years 2020 to 2022. Job listings with less than 100 views were excluded to make sure that the mean proportions are not biased by individual data points with a very low number of views (for example, the extreme case of a job listing with 1 male view and 0 female views, which would enter the analysis on a par with a listing with, say, 100 male and 0 female views). After filtering out all job listings with less than 100 views, our dataset contained 256,934 job listings that were viewed 47,937,792 times altogether (mean = 186.6, sd = 208.3). We did not consider any views by non-registered users of the platform, as we do not have any information about their gender identities. Also, views by registered users identifying as non-binary were not taken into account for the present study, as there are very few of them according to StepStone.

The data collection process was conducted by StepStone according to their rigorous code of conduct, and in compliance with GDPR requirements. It was approved by the company's legal team. The anonymity of all data points was maintained rigorously. To ensure maximum anonymity, the dataset provided to the authors of the present paper only contains aggregated data: We have no information about individual viewers but only the raw numbers of users that have viewed a specific job ad, along with their gender. Where available, the gender identity given in the individual user profile was taken into account; in all other cases, a gender guesser algorithm was used to gauge the user's gender. Importantly, this was done by StepStone using in-house solutions; no personal data (such as first names) were disclosed to the authors of the present paper. The methodological consequences and the limitations that this entails will be discussed in more detail below.

As the dataset is too large for hand coding the individual listings, they were grouped automatically according to the type of gender-sensitive phrasing that they use (called "gendering type" in the remainder of this paper, see Footnote 2). The most common variant is the GM in combination with an addition like *(m/w/d)* to indicate that the job position is open for people of male (m), female (w) or diverse (d) gender. The use of *(m/w/d)* is the de-facto standard since January 2019, when the German Civil Status Act (*Personenstandsgesetz*) was changed to allow for the gender entry *divers* 'diverse' in response to a verdict of the Federal Constitutional Court (Gesetz zur Änderung der in das Geburtenregister einzutragenden Angaben, last checked 27/06/2024). The other categories, listed in (2) to (8) below, were identified on the basis of a simple search for the relevant strings. If a job listing neither contained *(m/w/d)* nor any of the other relevant strings (such as the suffix <-in> or the asterisk <*>), it was not taken into account in our analysis as in such cases, we cannot be entirely sure whether it is a GM or a gender-neutral form that was not explicitly searched for.

The gendering types can be clustered into these groups:

1. Addition: GM combined with an orthographically accepted addition in the form of a phrase or abbreviations, mostly *(m/w/d)*, but other additions like *gn* (for *gender-neutral*) are possible: *Lehrer (m/w/d)*, *Lehrer (all genders)*, *Lehrer (gn)*

2. Single asterisk: a symbol that is not part of traditional orthography, serving the explicit purpose of including all genders, without adding the female suffix *-in*: *Lehrer\**

3. Neutral form with *-kraft* or participle: compounds with the last constituent *-kraft* (roughly: 'staff member') alternative forms to the GM that are not regarded as specified in respect to their semantic gender. In the case of participles, this only applies to plural forms: Lehrende (literally: 'teaching (persons)') is gender-neutral, Lehrender 'teaching (person) (m.)' is not.: *Lehrkraft*, *Lehrende*

4. Slash: a slash used between the stem and the female suffix as an orthographically accepted way of indicating both genders: *Lehrer/in*, *Lehrer/innen*

5. Capital I: use of a word-internal capital letter I to indicate that both men and women are included: *LehrerIn*, *LehrerInnen*

6. Morpheme separators: a symbol used between the stem and the female suffix: *Lehrer\*in*, *Lehrer_in*, *Lehrer:in*

7. Both forms: use of both the male and the female form: *Lehrerin oder Lehrer* 'teacher-f. or teacher-m.', *Lehrerinnen und Lehrer* 'teachers-f. and teachers-m.'

8. Only feminine form: *Lehrerin*, *Lehrerinnen*

Note that the groups differ in the extent to which they are seen as compatible with German orthographic conventions: gendering types using morpheme separators (such as *Lehrer\*innen*, *Lehrer:innen*) or sentence-internal majuscules (*LehrerInnen*) have caused the most controversy in public debates and parts of the academic literature, and they have been described as impractical and ideologically motivated. This is partly because the use of symbols like the underscore or the asterisk has been introduced with the intention of including non-binary individuals. Kotthoff [9 p107] therefore argues that their use indexes a specific ideological position.

Given that there are still many differences in the representation of women across different branches, we can expect the proportion of female views to be lower in traditionally male-dominated sectors. We have therefore divided the job listings into eight groups according to their affiliation with large job sectors, making use of the categories that StepStone uses for classifying their listings. In some cases, a job listing is associated with more than one sector. For the present analysis, we omit these cases in order to ensure that the influence of this variable can be factored in as reliably as possible. This leaves 685,583 listings that were taken into account in our analysis. Table 1 gives an overview of the total number of job titles viewed at least 100 times in each sector, along with the gendering type being used. Some of the job titles use multiple forms, e.g. *Lehrer\*in (m/w/d)*. In such cases, the first instance is taken into account, but we also annotated whether multiple forms were used. This means that *Lehrer (m/w/d)* would be counted as an instance of (1) addition (m/w/d) and tagged as using a single gendering type, while *Lehrer\*innen oder Dozierende (m/w/d)* (roghly: 'teachers or docents (m/w/d)') would be counted as an instance of (6) morpheme separator and tagged as using multiple gendering types. To disentangle the influence of gendering type and discipline, a generalised linear model was fit to the data using the package lme4 [32] for R [33]. The counts of female and male listing views were used as the response variable, and gendering type, discipline, as well as single vs. multiple occurrence of gendered forms were used as predictor variables.

## 4.2 Results

Fig 1 shows the relative proportion of female users in the total number of views of job listings in a specific discipline, broken down by gendering type. First of all, it immediately becomes clear that there are considerable differences between the different disciplines. As could probably be expected given traditional gender stereotypes, job listings in traditionally "male" fields of occupation such as the construction industry or IT and engineering are viewed more rarely by female users than listings in domains like administration and bookkeeping or the health sector. More relevant for our purposes, however, are the differences between gendering types. As mentioned above, the use of additions is by far the most frequent gendering type in our data. Hence, it is not surprising that the mean proportion of female views in this gendering type coincides almost perfectly with the mean proportion of overall female views in the respective sector, as indicated by the dashed lines in Fig 1. Interestingly, however, listings that make use of one of the other options, with a few exceptions, tend to show a higher proportion of

**Table 1. Job listings with at least 100 views across gendering types and sectors.**

| | Administration, Personnel, Bookkeeping | Construction industry, crafts | Finances | Health | IT, Engineering | Logistics, distribution & sales | Management | Service, counselling, marketing |
|---|---|---|---|---|---|---|---|---|
| **(1) addition (m/w/d)** | 49049 | 16936 | 9606 | 4013 | 33382 | 61362 | 23702 | 30411 |
| **(2) single asterisk** | 1099 | 206 | 189 | 59 | 646 | 896 | 414 | 629 |
| **(3) neutral form or participle** | 1000 | 769 | 78 | 308 | 116 | 1167 | 334 | 84 |
| **(4) slash /** | 2236 | 587 | 527 | 138 | 526 | 1908 | 529 | 670 |
| **(5) capital I** | 37 | 10 | 19 | 2 | 4 | 66 | 22 | 30 |
| **(6) morpheme separator (\*, \_,:)** | 3658 | 694 | 545 | 143 | 1037 | 1510 | 1093 | 1170 |
| **(7) both forms** | 8 | 0 | 2 | 0 | 1 | 4 | 7 | 2 |
| **(8) only feminine form** | 1519 | 275 | 162 | 42 | 273 | 362 | 408 | 253 |

female views. While caution is advised in the interpretation of the results (see section 4 below), there is an overall tendency for forms that make the female form explicit (using both forms, only the feminine form, or a morpheme separator between the stem and the suffix) to entail a higher proportion of female views, even though this effect is rather subtle in most cases. As we will discuss below, this effect becomes more pronounced when accounting for certain extralinguistic factors and taking a more detailed look into the data.

The coefficients of the generalised linear model are given in Table 2. All variance inflation factors are below 5, suggesting that the model does not suffer from multicollinearity. McFadden's $R^2$ is 0.46, indicating excellent model fit [34]. However, an important limitation of the model has to be mentioned: As mentioned in section 3, for privacy reasons, we do not have any data about individual usage profiles–in other words, we do not know which of the views that entered the model go back to the same individual. Thus, we do not know to what extent the data points are independent from one another, and we were unable to work with individual-specific random intercepts. However, given the large amount of data, we assume that the results of a mixed model with random effects for individuals would not have differed very strongly from the present one. An ANOVA comparing a null model with the full model shows that all three predictors emerge as highly significant. The effects plot in Fig 2 shows the model's predictions for the individual categories.

## 5. Discussion

Both the overview in Fig 1 and the results of the regression model reveal a number of interesting trends. However, we should first emphasise that all results have to be handled with caution as there are a number of extralinguistic factors that could not be taken into account, as we will explain in more detail below. Despite those limitations, the data show some tendencies indicating that the use of gendered forms does make a difference when it comes to attracting the attention of female users. First of all, the use of certain gendered forms is correlated with a higher proportion of female views in the majority of job sectors. Although there is considerable variation in the effect of different gendering types across different disciplines, as Fig 1 shows, the overall tendency is fairly clear: Variants that make the female form explicit on the linguistic surface are correlated with the highest proportion of female views, while the addition of *(m/w/d)* as in *Lehrer (m/w/d)* or the mere addition of an asterisk (*Lehrer\**) without adding a female suffix leads to the smallest proportion of female views.

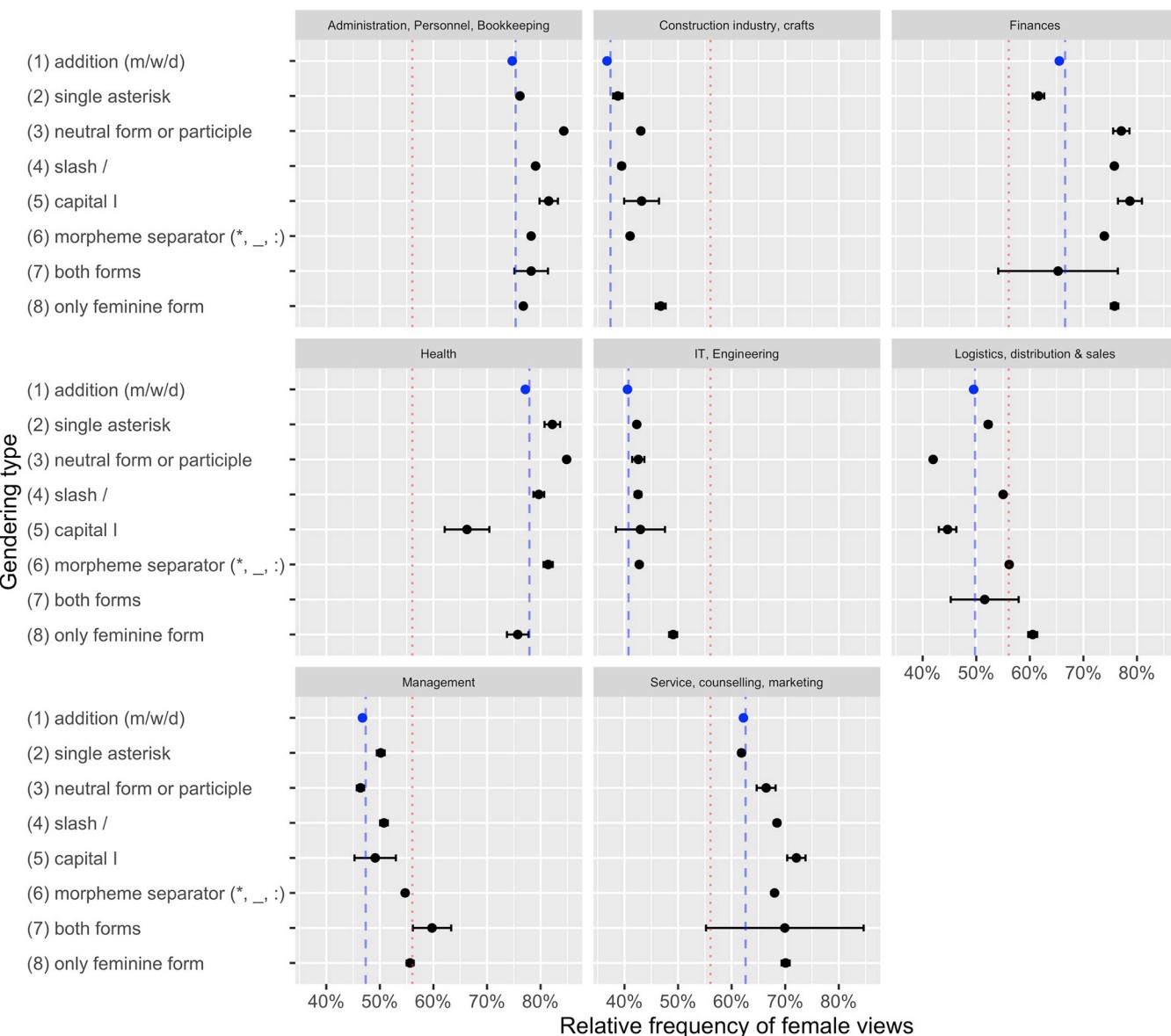

**Fig 1. Proportion of female views across different disciplines and different gendering types.** In each panel, the dashed blue line indicates the mean proportion of female views in the respective discipline, while the dotted red line indicates the overall mean proportion of female views across all disciplines. The bars indicate the standard error of the mean.

The overall results are consistent with many of the findings obtained in the previous literature. For one thing, it is striking that the use of only male forms, even if combined with *(m/w/d)* or a single asterisk, is fairly consistently correlated with a lower proportion in female views compared to other gendering types. This suggests that it is particularly the explicit female form with the feminine suffix *-in* that makes a difference here. To some extent, this is not surprising, as it does exactly what 'gendering' is supposed to do from the perspective of feminist linguistics, viz. making women visible in language [35 p82]. There are several possible explanations for the effect we observe: Women could be more inclined to view job listings that use these gendering types because they feel more targeted as the explicit use of the female form triggers a stronger mental representation of female persons, but it is also possible that they show a higher

**Table 2. Coefficients of a generalised linear model with gender (female/male) as response variable and gendering type and discipline as predictor variables.**

|  | Estimate | Std. Error | z value | Pr(>\|z\|) |
|---|---|---|---|---|
| (Intercept) | -0.53 | 0.00089 | -600 | <2e-16*** |
| gendering_type: (2) single asterisk | 0.074 | 0.0021 | 35.16 | <2e-16*** |
| gendering_type: (3) neutral form or participle | 0.12 | 0.0029 | 40.65 | <2e-16*** |
| gendering_type: (4) slash / | 0.19 | 0.0024 | 77.89 | <2e-16*** |
| gendering_type: (5) capital I | 0.13 | 0.0098 | 13.01 | <2e-16*** |
| gendering_type: (6) morpheme separator (*, _,:) | 0.21 | 0.0019 | 113.25 | <2e-16*** |
| gendering_type: (7) both forms | 0.25 | 0.024 | 10.43 | <2e-16*** |
| gendering_type: (8) only feminine form | 0.27 | 0.003 | 90.93 | <2e-16*** |
| discipline: IT, Engineering | 0.16 | 0.0011 | 143.12 | <2e-16*** |
| discipline: Management | 0.34 | 0.0011 | 301.79 | <2e-16*** |
| discipline: Logistics, distribution & sales | 0.47 | 0.001 | 468.59 | <2e-16*** |
| discipline: Service, counselling, marketing | 1.03 | 0.0011 | 917.84 | <2e-16*** |
| discipline: Finances | 1.12 | 0.0015 | 723.27 | <2e-16*** |
| discipline: Administration, Personnel, Bookkeeping | 1.62 | 0.0011 | 1516.5 | <2e-16*** |
| discipline: Health | 1.76 | 0.0021 | 820.06 | <2e-16*** |
| occurrences: multiple | 0.03 | 0.0021 | 14.55 | <2e-16*** |

degree of user interaction because they feel addressed more directly by a job listing that explicitly uses female forms. In the latter case, it would primarily be metalinguistic factors that drive the behavioural observations, while it would be primarily linguistic (more precisely: semantic) ones in the former. However, it is also possible that when advertising job types more targeted at women (perhaps because of gender stereotypes), companies might be more prone to use progressive types of gendering–for instance, using only the female form for a stereotypically female occupation like *Sekretärin* 'secretary (f.)' but gendered forms for less stereotypically gender-specific occupations like *Buchhalter*in* 'bookkeeper'. In this case, the effects we observe would have been brought about largely by extra-linguistic factors. While we cannot fully exclude this possibility because we work with aggregate data and do not have access to the individual job listings, it seems unlikely that this is the main factor influencing the results (we will discuss this possibility in more detail below). Instead, the most likely explanation is probably a combination of linguistic, meta-linguistic, and extra-linguistic factors that interact in intricate ways.

As such, it is no surprise that the results are not fully straightforward. In some sectors, the overall pattern is much less clear than in others. In the health sector, for instance, listings that use capital I for gendering as well as listings mentioning only the female form show a below-average proportion of female views. This is one of the cases where the aforementioned influence of extralinguistic factors plays an important role. More specifically, these observations can probably be explained by the types of jobs the different listing titles refer to, the current gender ratios in these jobs, and specific naming conventions for these jobs. For instance, there is a neutral term for nurses, *Pflegekraft*. We can expect that terms like *Pflegekraft* or *Pflege-fachkraft* (lit. 'nursing specialist') make up a large proportion of the neutral terms in this category. Indeed, a search for listings in the health sector with titles containing the string <pflege> or <Pflege> reveals that most nursing positions are gendered using *-kraft* (1320 out of 2208 jobs in the health sector containing the string <pflege> or <Pflege>). Thus, the observation that neutral forms are correlated with a particularly high proportion of female views in the health sector can probably be explained by the fact that most of them are nursing positions, which still is a strongly female-dominated occupation. There is no established neutral term for

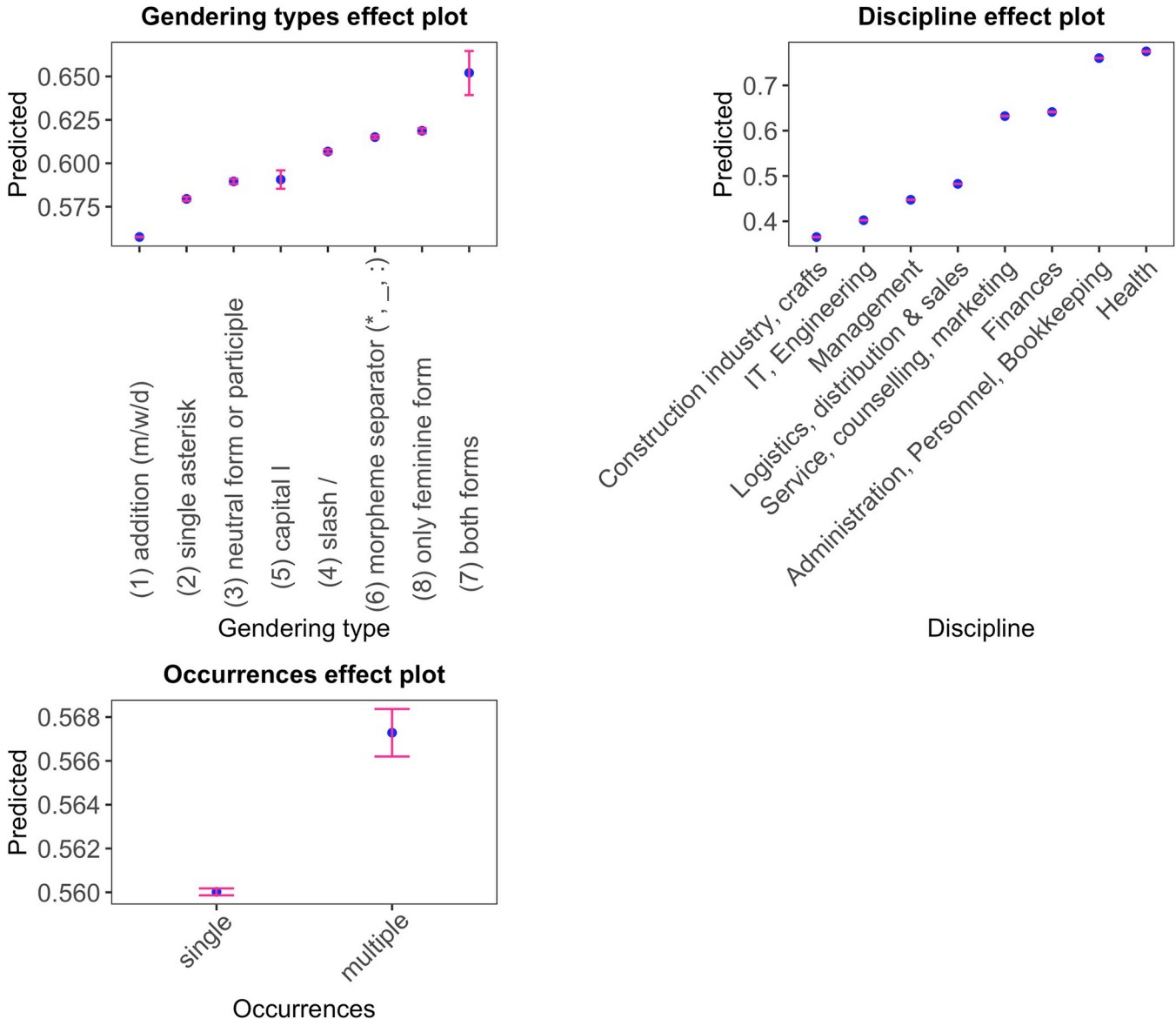

**Fig 2. Effect plot of gendering type, discipline, and single vs. multiple occurrence of gendered forms.**

physicians, by contrast, who are called *Arzt* (male) or *Ärztin* (female). Thus, we can expect that job listings for other, less female-dominated jobs make use of other gendering types.

This is consistent with other sectors in which certain positions can be gendered in a similar way, indicating that employers tend to use neutral forms when possible, rather than controversial gendering types (such as gendering with asterisk or colon, which are at the center of the heated public discussion about gender-sensitive language) or gender-specific traditional forms (like *Krankenschwester*, which used to be, and to some extent still is, the conventional term for nurses in German). Similar observations can be made in the logistics sector, in which the majority of jobs positions available are called *Fachkraft* (e.g. *Fachkraft für Lageristik* instead of the more traditional male form *Lagerist*). In contrast to the health sector, these jobs are traditionally occupied by men, leading to the opposite correlation: The neutral form correlates with the smallest proportion of female views.

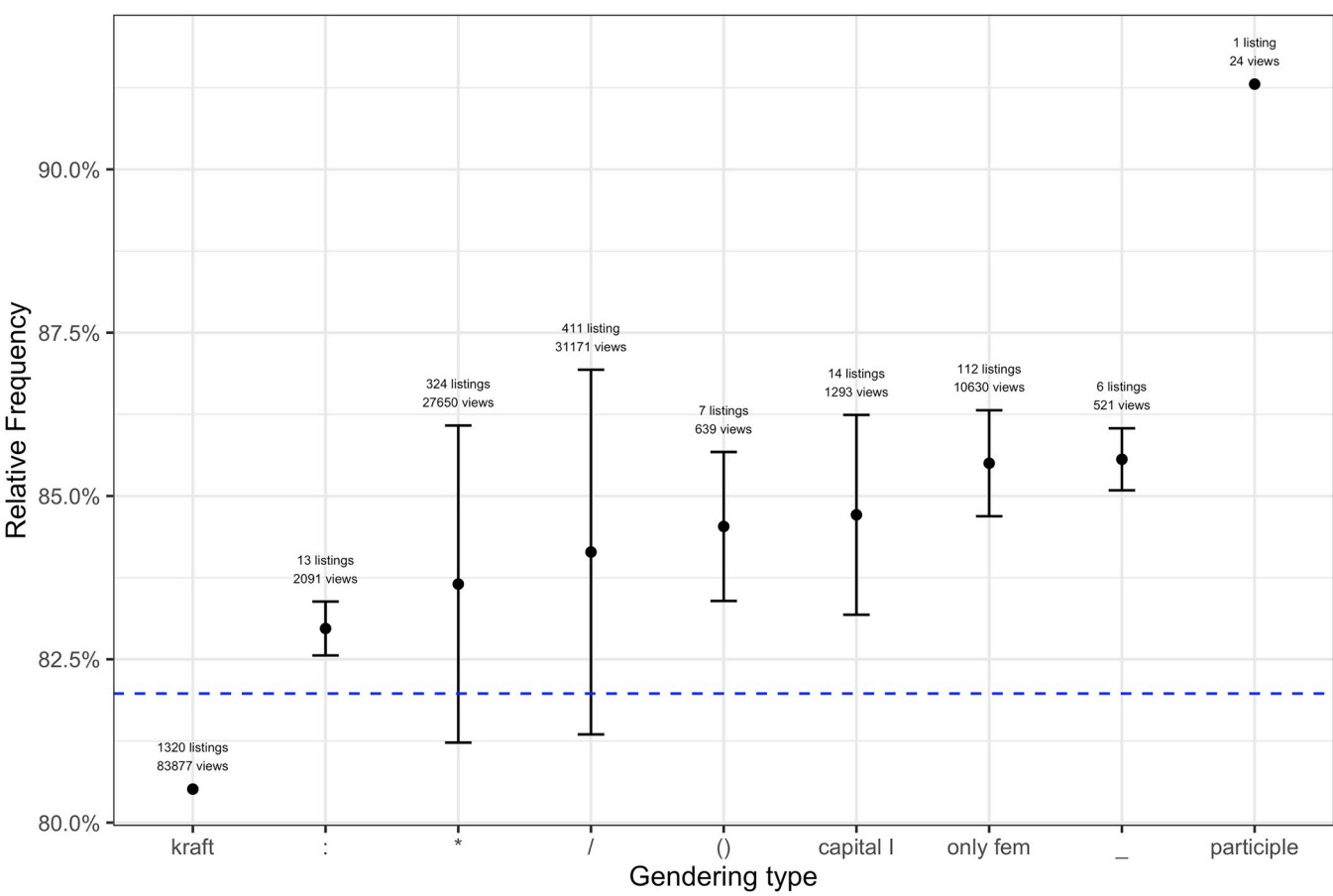

**Fig 3. Proportion of female views for listings in the health sector starting with the string <Pflege-/pflege->.** The dashed blue line shows the overall mean proportion of female viewers across all listings starting with <Pflege-/pflege->.

To minimise similar extralinguistic effects, we have inspected the different gendering types in job titles for a single profession, comparing only job listings for nurses. Doing so may account for the effect observed above, revealing a pattern that is very similar to the overall tendencies mentioned: The neutral form (*Pflegekraft*) correlates with the smallest number of female views, while other, more explicit gendering types correlate with larger proportions, as can be seen in Fig 3. This suggests that even in a strongly female-dominated field like nursing, using gender-sensitive language in job listings can have an effect on potential applicants.

In general, there are relatively strong differences between the individual gendering types. Even the forms using morpheme separators differ to a considerable extent. As mentioned above, the ones that appear to attract the largest female viewership in one way or another all include the feminine suffix *-in*. Interestingly, the use of a single asterisk, as in *Lehrer\** 'teacher\*', does not tend to lead to an increase in the proportion of female views. This is interesting in the light of the question of how effective different gendering types are, especially given that the asterisk has become the epitome of gender-sensitive language in the public debate [36]. Kotthoff [9 p116] states that the asterisk is intentionally used to signal the producers' position on certain social and moral issues and to place them in a certain socio-political group. Given the broad public discussion, we can assume that language users are metalinguistically aware of this type of signalling. Still, this presumed metalinguistic awareness does not translate into a higher proportion of female views. A possible explanation for this finding is

that the intended meaning of the asterisk does not override the male associations of the stem form. If this interpretation is correct, it lends further support to previous findings in psycholinguistic research suggesting that masculine forms are strongly associated with male representations.

This is relevant to the question of whether our results can partly be attributed to some of the extralinguistic factors that have been used to question the conclusions of previous studies in the existing literature. At the end of section 3, we have mentioned what can be seen as major potential problems of previous empirical approaches. Our study of real-life user interactions arguably escapes some of them: Firstly, our study does not face the problem that its results could be explained by the effects of mismatching relative pronouns or other syntactic aspects. The titles of job listings usually consist of simple noun phrases, which means that no larger syntactic structures can be found that could influence the results. Secondly, experimental approaches often suffer from the fact that participants can be biased in one way or another due to the laboratory setting. This is not the case here as we have used real-life data that represent genuine reactions to real-life language in a relevant social context. Thirdly, experimental studies often draw on small, non-representative samples. Our study has a very large sample size, although it is limited to registered users of the job platform StepStone. Nevertheless, there are important limitations that need to be acknowledged. Apart from the fact that StepStone users may not be a fully representative sample, we have to draw on aggregated data, which means that we do not have access to potentially relevant information such as the exact wording of the job titles. The reasons why users do or do not interact with a job listing are complex and manifold, and so are the reasons why employers choose specific forms of gender-sensitive language–as mentioned above, we therefore have to be very careful with any causal interpretation of the results of the present study, and more detailed follow-up studies are necessary to back up our tentative conclusions. In section 6, we discuss some possibilities for future avenues of research.

## 6. Conclusion and outlook

Drawing on data of real-life interaction with job listings, we have investigated the effects of different types of gender-sensitive language ("gendering"). Our results show that across different job sectors, some types of gender-sensitive language correlate with higher proportions of views by female users of the platform. This is true for individual jobs as well, as the comparison of different job titles for nursing positions has shown: Even in a single strongly female-dominated profession, gendering job titles in a specific fashion appears to attract more female viewers.

Many of the effects that we have observed can be accounted for by extralinguistic factors, some of which we have discussed above. Future studies using similar approaches will have to try to account for as many of these extralinguistic factors as possible. Intriguingly, when taking those extralinguistic effects into account, the overall effect seems to become even clearer, as we can easily explain outliers such as the comparatively low proportion of female viewers for job listings using a capital I or only the feminine form in the health sector.

Probably the most relevant result of our study is that the proportion of female viewers (with very few exceptions that can likely be explained by extralinguistic factors) is consistently higher when types of gender-sensitive language are used that explicitly mention the feminine form with the suffix -in. This aligns well with some findings of experimental studies, as well as with ideas common in parts of the feminist-linguistic literature aiming at making women 'visible' on the linguistic surface. The use of so-called generic masculine forms with an addition like (m/w/d) correlates with the smallest proportion of views by users who identify as female. A very similar observation can be made with respect to neutral forms and the masculine form

accompanied by a single asterisk (even though the so-called gender star $<^*>$ is described as a prominent symbol of gender-sensitive language and feminism in corpus-based studies, which is why one could expect readers to associate it with gender diversity).

The specific limitations and advantages of our approach differ from those of previous studies on the subject. The analysis of online interaction avoids some of the disadvantages of corpus-based and experimental approaches by analysing real-life reactions to real-life language, which is why our approach offers a more direct window to the behavioural correlates of different gendering types. At the same time, it only allows for conclusions regarding mental representations or semantics of the expressions in question to a limited extent, because there are a number of important extralinguistic factors at play.

For a more detailed explanation of the observed patterns, big-data approaches like ours need to be complemented by others, particularly experimental ones. While our results are compatible with previous research suggesting that the so-called generic masculine is not consistently interpreted in a gender-neutral way, further research on the topic is needed, not only to better explain our results and those of previous small-scale studies, but also to find ways of reconciling different methodologies. For example, following up on the small case study zooming in on job titles in the health sector presented in section 5, it would be worthwhile to investigate specific job titles in more detail, taking the exact wordings of the relevant titles into account. Also, it would be interesting to conduct an experimental study in which participants see different versions of the same job title with different gendering types.

Gender-sensitive language is a complex issue, not only in linguistics, but also in the political sphere, where a variety of social, political and ethical factors have to be taken into account. Linguistic research like ours can only answer some of the many questions relevant to decisions on gender-sensitive language, which is why we refrain from discussing any potential implications of our results for linguistic policies (e.g., guidelines for gender-sensitive language).

All in all, we hope to have shown that the analysis of online user interaction data can contribute valuable insights to the study of gender-sensitive language, and we believe that this approach also has much potential for addressing follow-up questions in future research.

## Acknowledgments

We are grateful to StepStone for providing the data for the present analysis, and particularly to Tanja Winkler and Timm Lochmann for the good and constructive collaboration and for their enthusiasm for this project. We extend our gratitude to Marlene Rummel, who has provided us with invaluable expertise and assistance, and to the anonymous reviewers for their feedback that has helped to make the paper more focused.

## Author Contributions

**Conceptualization:** Dominik Hetjens, Stefan Hartmann.

**Data curation:** Dominik Hetjens, Stefan Hartmann.

**Formal analysis:** Dominik Hetjens, Stefan Hartmann.

**Investigation:** Dominik Hetjens, Stefan Hartmann.

**Methodology:** Dominik Hetjens, Stefan Hartmann.

**Project administration:** Dominik Hetjens, Stefan Hartmann.

**Resources:** Dominik Hetjens, Stefan Hartmann.

**Software:** Dominik Hetjens, Stefan Hartmann.

**Visualization:** Dominik Hetjens, Stefan Hartmann.

**Writing – original draft:** Dominik Hetjens, Stefan Hartmann.

**Writing – review & editing:** Dominik Hetjens, Stefan Hartmann.

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
