## [Decision Letter · Decision Letter 0]

14 May 2024

PONE-D-24-04132Effects of gender sensitive language in job listings: a study on real-life user interactionPLOS ONE

Dear Dr. Hetjens,

Thank you for submitting your manuscript to PLOS ONE. Your manuscript has been reviewed, and the comments of the reviewers are included at the bottom of this letter. Both are positive and recommend its publication pending minor reviews. Therefore, we invite you to submit a revised version of the manuscript that addresses the points raised during the review process.

We look forward to receiving your revised manuscript.

Kind regards,

Montserrat Comesaña Vila

Academic Editor

PLOS ONE

Journal Requirements:

2. In your Methods section, please include additional information about your dataset and ensure that you have included a statement specifying whether the collection and analysis method complied with the terms and conditions for the source of the data.

Reviewers' comments:

Reviewer's Responses to Questions

**Comments to the Author**

1. Is the manuscript technically sound, and do the data support the conclusions?

Reviewer #1: Yes

Reviewer #2: Yes

2. Has the statistical analysis been performed appropriately and rigorously? 

Reviewer #1: Yes

Reviewer #2: Yes

3. Have the authors made all data underlying the findings in their manuscript fully available?

Reviewer #1: Yes

Reviewer #2: Yes

4. Is the manuscript presented in an intelligible fashion and written in standard English?

Reviewer #1: Yes

Reviewer #2: Yes

5. Review Comments to the Author

Reviewer #1: The entire manuscript is exceptionally well-written, featuring a meticulously structured case study section, clear objectives, and a robust methodology. The study demonstrates a strong motivation, drawing from previous research, and the questions posed within this manuscript are pertinent. They have the potential to yield significant insights into the impact of gender-sensitive language, particularly in the context of job listings, which is an important and timely topic. This paper is poised to make substantial contributions across various fields, with the potential for significant impact. Moreover, it introduces a novel perspective by analyzing real language usage with a large sample size to explore gender-sensitive language, while also providing a methodology that can pave the way for further research inquiries. However, navigating through the manuscript, especially the introduction, can prove challenging. I believe that reorganizing and simplifying the content is necessary to facilitate comprehension for a wider audience.

I recommend that the author begins with a more general review of gender and gender sensitivity, along with their implications, before delving into the specific situation in German language. The current introduction may deter some readers, as it appears overly focused on the German context, whereas I believe it holds broader implications. While I appreciate the authors' thorough introduction, I suggest trimming it down and focusing on more specific points to avoid repetition. For instance, the method of specifying the feminine is mentioned multiple times through the addition of suffixes. Additionally, some of the reviewed studies contain excessive detail, such as including the professions of the authors, which may fatigue external readers. The introduction would benefit from being streamlined and more concise overall. It currently resembles a chapter from a book rather than an introduction to a paper.

The subsections on "Systemic Approach" and "Corpus-based Approach" incorporate various studies, theoretical frameworks, and underlying philosophies of each approach. While it's important to explain these to readers, they don't have a direct link to the main topic. However, the "Experimental Approach" includes specific studies relevant to job advertisements, which is the primary focus. It might be more effective for the author to begin with literature directly related to job advertisements and the experimental approach. Currently, readers have to wait until page 22 to encounter specific literature on the main topic. I recommend introducing this literature on job advertisements earlier, followed by a brief overview of how systemic and corpus-based approaches can aid in investigating this topic, emphasizing their relevance to gender studies. Each approach could be presented in a condensed and more focused manner. Additionally, since the study combines corpus-based and experimental approaches, less emphasis should be placed on systemic approaches.

In the case study section, it's challenging to discern which aspects of the current study and the analysis the authors consider part of the experimental approach versus the corpus-based approach. Since the main objective of the study is to capitalize on the strengths of both methodologies, it's crucial to clarify this explicitly. While it's apparent that classifying job listings based on genderism within the corpus analysis, and tracking the number of views by real users aligns more with the experimental approach, the distinction isn't sufficiently clear. This becomes especially important given the absence of additional data such as reaction times, acceptability tests, or eye-tracking, which are typically integral components of the experimental approach. Therefore, I suggest strengthening the separation between the two methodologies to provide greater clarity and understanding for readers.

Reviewer #2: The study investigated the effects of gender-sensitive language (in German, which is a language with three grammatical genders masculine, feminine and neuter) in job advertisements on user interaction. It gathered anonymous data on user interactions, specifically click counts, from a company's website, employing a novel approach that blends corpus-based and experimental methods. Strengths of the paper include its innovative use of real-world interaction data and its timely focus on gender-sensitive language in job postings. I also commend the authors for their transparent methodological reporting.

However, the study has some methodological limitations, most of which are aknowledged by the authors. It draws data from a specific recruitment platform, potentially introducing biases based on user demographics and behavior, thus limiting generalizability. The study recognizes extralinguistic factors like job sector, location, and company reputation, which are challenging to control in a naturalistic setting. While the study identifies correlations between gender-sensitive language and user behavior, it cannot definitively establish causation, as other unmeasured variables may influence observed patterns. Acknowledging these limitations, the authors suggest avenues for future research to enhance the generalizability and robustness of findings, including diversifying datasets and contexts.

As such, I think the authors may find this recent paper on ideologies relevant, as it delves into gender-inclusive language in German across diverse user groups, including those who learned German as adults exclusively and German speakers residing outside German-speaking regions.

Truan, Naomi. "Whose language counts?: Native speakerism and monolingual bias in language ideological research: Challenges and directions for further research" European Journal of Applied Linguistics, 2024. https://doi-org.ezproxy.leidenuniv.nl/10.1515/eujal-2024-0006

Additionally, considering multilingual societies where speakers often default to either the masculine or feminine gender when code-switching languages could offer valuable insights about gender inclusiveness. For instance, Balam (2016) and Valdés Kroff (2016) discuss examples from Spanish/English in Miami and Belize (' el.masc Virgin Mary' ) at a rate between 97 and 99% of all switched nominal constructions. Parafita Couto et al. (2015) provide examples of a feminine deafault from Spanish-Basque ('la.fem osaba' - ' the uncle') . Bellamy and Parafita Couto (2022) offer an overview of such strategies. Reflecting on these phenomena across diverse communities, where multilingual speakers navigate languages with varying gender categories, could significantly enhance future studies on gender sensitivity and gender-inclusive language within these contexts.

Balam, O. (2016) Semantic categories and gender assignment in contact Spanish: Type of code-switching and its relevance to linguistic outcomes. Journal of Language Contact, 9, 405–435.

Bellamy, K and Parafita Couto, MC (2022) Gender assignment in mixed noun phases: State of the art. In Ayoun, D (ed), The Acquisition of Gender: Crosslinguistic perspectives. Amsterdam: John Benjamins. pp. 13–48. https://doi.org/10.1075/sibil.63.02bel

Parafita Couto, M. C., Munarriz, A., Epelde, I., Deuchar, M., & Oyharçabal, B. (2015) Gender conflict resolution in Spanish-Basque mixed DPs. Bilingualism: Language and Cognition, 18(2), 304–323.

Valdés Kroff, J. R. (2016) Mixed NPs in Spanish-English bilingual speech: Using a corpus-based approach to inform models of sentence processing. In R. E. Guzzardo Tamargo, C. M. Mazak & M. C. Parafita Couto (Eds.), Spanish-English code-switching in the Caribbean and the US (pp. 281–300). John Benjamins.

6. PLOS authors have the option to publish the peer review history of their article (what does this mean?). If published, this will include your full peer review and any attached files.

Reviewer #1: **Yes: **Antonio Iniesta

Reviewer #2: No

---

## [Author Response · Author response to Decision Letter 0]

4 Jul 2024

Response to reviewers

We would like to thank the reviewers for their helpful and thorough feedback. Following the reviewers’ advice, we have restructured the paper and contextualized the discussion of gender-sensitive language much more. While we have tried to incorporate all suggestions put forward by Reviewer 1, some of Reviewer 2’s suggestions go beyond the scope of our paper, although they are definitely very interesting and relevant. As we are focusing on the empirical question of the potential effects of gender-sensitive language on user interactions with online job advertisements, we cannot say much about potential effects of multilingualism on the use of grammatical genders. However, we agree that this is an aspect that should be pursued in future research, and we have added some ideas to the outlook. 

Below we respond to the individual issues raised by the reviewers in more detail.

Reviewer #1: 

Point 1.1

The entire manuscript is exceptionally well-written, featuring a meticulously structured case study section, clear objectives, and a robust methodology. The study demonstrates a strong motivation, drawing from previous research, and the questions posed within this manuscript are pertinent. They have the potential to yield significant insights into the impact of gender-sensitive language, particularly in the context of job listings, which is an important and timely topic. This paper is poised to make substantial contributions across various fields, with the potential for significant impact. Moreover, it introduces a novel perspective by analyzing real language usage with a large sample size to explore gender-sensitive language, while also providing a methodology that can pave the way for further research inquiries. However, navigating through the manuscript, especially the introduction, can prove challenging. I believe that reorganizing and simplifying the content is necessary to facilitate comprehension for a wider audience.

Reply 1.1

Thanks for the positive assessment and for the constructive feedback! We hope that our revisions sufficiently address the very legitimate concerns regarding the readability and accessibility of the paper. In particular, we have considerably shortened and condensed the introduction as well as the research overview, and contextualized the specific discussion more in a general context.

Point 1.2

I recommend that the author begins with a more general review of gender and gender sensitivity, along with their implications, before delving into the specific situation in German language. The current introduction may deter some readers, as it appears overly focused on the German context, whereas I believe it holds broader implications. 

Reply 1.2

This is indeed a very relevant aspect. We have added a short discussion about language and gender more generally, abstracting away from the specifics of German. At the same time, we have refrained from providing an in-depth overview of the (linguistic and/or public) debates on language and gender, as this topic is not only too broad to be adequately discussed in such a short paper, but addressing it in too much detail might also invite the assumption that we are overestimating the implications of our study.

Point 1.3

While I appreciate the authors' thorough introduction, I suggest trimming it down and focusing on more specific points to avoid repetition. For instance, the method of specifying the feminine is mentioned multiple times through the addition of suffixes. Additionally, some of the reviewed studies contain excessive detail, such as including the professions of the authors, which may fatigue external readers. The introduction would benefit from being streamlined and more concise overall. It currently resembles a chapter from a book rather than an introduction to a paper.

Reply 1.3

We have restructured the paper to keep the introduction significantly shorter, and we have followed the advice to trim down the literature review. In particular, we have substantially shortened the overview of corpus-based approaches, as they are not as relevant to our own study as previous experimental work on the topic. To still provide some of the information included in the original introduction, we have added a short additional section between the introduction and the research overview, in which we provide a short overview of grammatical genders in German. This way, we intend to keep the introduction short and more concise, while at the same time providing the most important information necessary to understand the study. 

Point 1.4

The subsections on "Systemic Approach" and "Corpus-based Approach" incorporate various studies, theoretical frameworks, and underlying philosophies of each approach. While it's important to explain these to readers, they don't have a direct link to the main topic. However, the "Experimental Approach" includes specific studies relevant to job advertisements, which is the primary focus. It might be more effective for the author to begin with literature directly related to job advertisements and the experimental approach. Currently, readers have to wait until page 22 to encounter specific literature on the main topic. I recommend introducing this literature on job advertisements earlier, followed by a brief overview of how systemic and corpus-based approaches can aid in investigating this topic, emphasizing their relevance to gender studies. Each approach could be presented in a condensed and more focused manner. Additionally, since the study combines corpus-based and experimental approaches, less emphasis should be placed on systemic approaches.

Reply 1.4

To address these points, we have condensed the research overview into one section, mentioning the systemic approach only briefly, and introducing the more relevant experimental studies earlier. The whole section now focuses strongly on research that has a more direct link to the main topic.

Point 1.5

In the case study section, it's challenging to discern which aspects of the current study and the analysis the authors consider part of the experimental approach versus the corpus-based approach. Since the main objective of the study is to capitalize on the strengths of both methodologies, it's crucial to clarify this explicitly. While it's apparent that classifying job listings based on genderism within the corpus analysis, and tracking the number of views by real users aligns more with the experimental approach, the distinction isn't sufficiently clear. This becomes especially important given the absence of additional data such as reaction times, acceptability tests, or eye-tracking, which are typically integral components of the experimental approach. Therefore, I suggest strengthening the separation between the two methodologies to provide greater clarity and understanding for readers.

Reply 1.5

We have added a few remarks to make it more clear in which way parts of our study can be considered part of the corpus-based approach. Overall, we believe that our approach is - to the best of our knowledge - new, and cannot be classified according to the classification of approaches we have introduced in the research overview.

Reviewer #2: 

Point 2.1

The study investigated the effects of gender-sensitive language (in German, which is a language with three grammatical genders masculine, feminine and neuter) in job advertisements on user interaction. It gathered anonymous data on user interactions, specifically click counts, from a company's website, employing a novel approach that blends corpus-based and experimental methods. Strengths of the paper include its innovative use of real-world interaction data and its timely focus on gender-sensitive language in job postings. I also commend the authors for their transparent methodological reporting.

However, the study has some methodological limitations, most of which are acknowledged by the authors. It draws data from a specific recruitment platform, potentially introducing biases based on user demographics and behavior, thus limiting generalizability. The study recognizes extralinguistic factors like job sector, location, and company reputation, which are challenging to control in a naturalistic setting. While the study identifies correlations between gender-sensitive language and user behavior, it cannot definitively establish causation, as other unmeasured variables may influence observed patterns. Acknowledging these limitations, the authors suggest avenues for future research to enhance the generalizability and robustness of findings, including diversifying datasets and contexts.

Reply 2.1

Thanks for the positive assessment!

Point 2.2

As such, I think the authors may find this recent paper on ideologies relevant, as it delves into gender-inclusive language in German across diverse user groups, including those who learned German as adults exclusively and German speakers residing outside German-speaking regions.

Truan, Naomi. "Whose language counts?: Native speakerism and monolingual bias in language ideological research: Challenges and directions for further research" European Journal of Applied Linguistics, 2024. https://doi-org.ezproxy.leidenuniv.nl/10.1515/eujal-2024-0006

Additionally, considering multilingual societies where speakers often default to either the masculine or feminine gender when code-switching languages could offer valuable insights about gender inclusiveness. For instance, Balam (2016) and Valdés Kroff (2016) discuss examples from Spanish/English in Miami and Belize (' el.masc Virgin Mary' ) at a rate between 97 and 99% of all switched nominal constructions. Parafita Couto et al. (2015) provide examples of a feminine deafault from Spanish-Basque ('la.fem osaba' - ' the uncle') . Bellamy and Parafita Couto (2022) offer an overview of such strategies. Reflecting on these phenomena across diverse communities, where multilingual speakers navigate languages with varying gender categories, could significantly enhance future studies on gender sensitivity and gender-inclusive language within these contexts.

Balam, O. (2016) Semantic categories and gender assignment in contact Spanish: Type of code-switching and its relevance to linguistic outcomes. Journal of Language Contact, 9, 405–435.

Bellamy, K and Parafita Couto, MC (2022) Gender assignment in mixed noun phases: State of the art. In Ayoun, D (ed), The Acquisition of Gender: Crosslinguistic perspectives. Amsterdam: John Benjamins. pp. 13–48. https://doi.org/10.1075/sibil.63.02bel

Parafita Couto, M. C., Munarriz, A., Epelde, I., Deuchar, M., & Oyharçabal, B. (2015) Gender conflict resolution in Spanish-Basque mixed DPs. Bilingualism: Language and Cognition, 18(2), 304–323.

Valdés Kroff, J. R. (2016) Mixed NPs in Spanish-English bilingual speech: Using a corpus-based approach to inform models of sentence processing. In R. E. Guzzardo Tamargo, C. M. Mazak & M. C. Parafita Couto (Eds.), Spanish-English code-switching in the Caribbean and the US (pp. 281–300). John Benjamins.

Reply 2.2

Thanks for these suggestions! They are very much in line with the first reviewer’s proposal to delve a bit more deeply into the issue of gender-sensitive language beyond the specifically German-language context. While the issue of multilingualism goes beyond the scope of the present paper, we have added a more thorough theoretical discussion that contextualizes our study in the more general debates about gender-sensitive language and language ideologies.

---

## [Editor Report · Decision Letter 1]

17 Jul 2024

Effects of gender sensitive language in job listings: a study on real-life user interaction

PONE-D-24-04132R1

Dear Dr. Dominik Hetjens,

We’re pleased to inform you that your manuscript has been judged scientifically suitable for publication and will be formally accepted for publication once it meets all outstanding technical requirements.

Kind regards,

Montserrat Comesaña Vila

Academic Editor

PLOS ONE

---

## [Editor Report · Acceptance letter]

22 Jul 2024

PONE-D-24-04132R1 

PLOS ONE

Dear Dr. Hetjens, 

I'm pleased to inform you that your manuscript has been deemed suitable for publication in PLOS ONE. Congratulations! Your manuscript is now being handed over to our production team.

Kind regards, 

on behalf of

Dr. Montserrat Comesaña Vila 

Academic Editor

PLOS ONE